# Pruning with Output Error Minimization for Producing Efficient Neural Networks

## Abstract

Deep Neural Networks (DNNs) are dominant in the field of machine learning. However, because DNN models have large computational complexity, implementation with resource-limited equipment is challenging. Therefore, techniques of compressing DNN models without degrading their accuracy is desired. Pruning is one such technique to remove redundant neurons (or channels). In this paper, we present Pruning with Output Error Minimization (POEM), a method that performs not only pruning but also *reconstruction* to compensate the error caused by pruning. The strength of POEM lies in its reconstruction to minimize the output error of the activation function, while the previous methods minimize the error of the value before applying the activation function. The experiments with well-known DNN models (VGG-16, ResNet-18, MobileNet) and image recognition datasets (ImageNet, CUB-200-2011) were conducted. The results show that POEM significantly outperformed the previous methods in maintaining the accuracy of the compressed models.

## 1 Introduction

Nowadays, Deep Neural Networks (DNNs) are dominant in the field of machine learning. The demand for DNNs is increasing in various applications. However, DNNs are known to be over-parameterized and require large computational cost. This makes them computationally slow, power-consuming, and difficult to be implemented in resource-limited equipment.

Therefore, there is a need for the techniques to create efficient DNN models by compressing large models while maintaining the accuracy. Pruning is one such technique to remove redundant weights from trained DNN models. Pruning methods can be divided into two groups: unstructured pruning and structured pruning. The former removes weight parameters in order to make the weight tensor sparse. Since the shape of the weight tensor remains the same, the compressed model should be implemented using hardware and libraries that can perform calculations only on non-zero weights. The latter removes neurons (or channels) in order to make the shape of the weight tensor smaller. Therefore, the effect of compression can be achieved by implementing the compressed model using general hardware and libraries. In this paper, we focus on structured pruning.

How well a pruned model maintains its accuracy depends on two factors. The first is compression ratio optimization, in other words, how many neurons are reduced in each layer. The other is layer-wise optimization, in other words, which neurons to be preserved in each layer.

In recent years, there is a growing awareness that the value of pruning lies in the search for an efficient sub-architecture out of a large redundant architecture. This is due to the research results showing that a DNN model with the pruned architecture trained from scratch can achieve at least as good accuracy as the pruned and fine-tuned model (Liu et al., 2019). For this reason, the recent trend is to focus on compression ratio optimization problem.

However, does it mean that the layer-wise optimization is no more important? It is reasonable to claim that combining a compression ratio optimization method with a better layer-wise optimization method should result in more effective pruning. Therefore, layer-wise optimization is still important and worth investigating.

In this paper, we propose a pruning method named *Pruning with Output Error Minimization* (POEM) that performs layer-wise optimization. The strength of POEM lies in its reconstruction using the

*Weighted Least Squares* (WLS) method so as to minimize the output error of the activation function, while the previous methods (Luo et al., 2017; He et al., 2017; Dong et al., 2017; Kamma & Wada, 2021) minimize the error of the value before applying the activation function. For example, since the ReLU function rounds a negative value to zero, the error on a negative element need not be compensated (unless it turns positive due to the error). POEM can perform reconstruction only for the positive elements, while the previous methods perform reconstruction for all elements including negative ones. For this reason, POEM is superior to the previous methods in maintaining the accuracy of the pruned model. To the best of our knowledge, POEM is the first method to perform reconstruction based on the output error of the activation function.

For verifying POEM, we conducted experiments on ImageNet (Deng et al., 2009), a large-scale image classification dataset, and well-known DNN models for image classification, such as VGG-16 (Simonyan & Zisserman, 2015), ResNet-18 (He et al., 2016), and MobileNet (Howard et al., 2017). The results show that POEM can prevent the output error of the activation function better than the previous methods (He et al., 2017; Kamma & Wada, 2021), and improve the accuracy both before and after fine-tuning. We also confirmed that the accuracy of the compressed model can be further improved by combining POEM and the compression ratio optimization methods (He et al., 2018b; Kamma et al., 2022; Li et al., 2022).

The rest of this paper is structured as follows. In Sec. 2, we introduce related works. In Sec. 3, we explain our proposed method. In Sec. 4, we show experimental results to verify the effectiveness of POEM. In Sec. 5, we conclude discussions in this paper.

## 2 RELATED WORKS

DNN compression methods can be divided into four groups: structured pruning, unstructured pruning (or sparsification), tensor decomposition, and quantization. Structured pruning is to make the shape of the weight tensor smaller by removing redundant neurons (or channels) (Molchanov et al., 2017; He et al., 2018a; 2017; Luo et al., 2017; Kamma & Wada, 2021; Jiang et al., 2018). The advantage of these methods is that the effect of compression can be obtained without special hardware or libraries. Unstructured pruning is to make the weight tensor sparse without changing its shape (LeCun et al., 1990; Liu et al., 2015; Han et al., 2016; Lee et al., 2019). The effect of unstructured pruning can be obtained by implementing the compressed model using hardware or libraries that can perform computation only for the non-zero elements of the weight tensor. The methods based on tensor decomposition replace a large weight tensor by the product of multiple smaller weight tensors (Xue et al., 2013; Kim et al., 2019; Denton et al., 2014). These methods can effectively reduce the number of parameters and the computational complexity, although the compressed model gets extra layers incurring an additional computational overhead. Quantization is to reduce the memory and complexity requirements of a model by discretizing the weights (Courbariaux et al., 2015; Liu et al., 2022; Li et al., 2021; Wei et al., 2022). A quantized model needs to be implemented on low-bit computation equipment.

In this paper, we focus on structured pruning because of the following benefits: the structured pruning methods can compress the model without incurring computational overhead; the compressed model can be implemented without special hardware or libraries.

For developing an effective pruning method, 2 problems should be addressed. One is the problem of compression ratio optimization, and the other is layer-wise optimization. Some pruning methods address both of these problems in a single framework, while others handle each problem separately.

The compression ratio optimization is to configure the number of pruned neurons in each layer. AutoML Model Compression (AMC) uses reinforcement learning to tune compression ratios so that the accuracy of the model is maximized posing a constraint on FLOPs (the number of floating point multiplications), or the FLOPs are minimized posing a constraint on accuracy (He et al., 2018b). Pruning Ratio Optimizer (PRO) tunes compression ratios by alternately performing layer selection and pruning based on the output error of the final layer (Kamma et al., 2022). RandomPruning performs random search in the search space of compression ratios (Li et al., 2022). These methods can be combined with any layer-wise optimization methods.

The layer-wise optimization is to select which neurons to be preserved in each layer. A lot of neuron selection criteria have been investigated, such as the ones based on the norm of outgoing

weights (He et al., 2014), the derivative information of the loss function (Molchanov et al., 2017), the geometric median of incoming weights (He et al., 2019), the output difference of final layer of the model (Yu et al., 2018; Luo & Wu, 2020), and so on. It is known that performing *reconstruction* to tune the remaining weights is effective for compensating the error caused by pruning. Channel Pruning (CP) (He et al., 2017), ThiNet (Luo et al., 2017), and Reconstruction Error Aware Pruning (REAP) (Kamma & Wada, 2021) use the Least Squares (LS) method for reconstruction. Layer-wise Optimal Brain Surgeon (LWOBS) (Dong et al., 2017) performs reconstruction based on the second order derivative information of the layer-wise Mean Squared Error (MSE). With these methods, a high compression ratio can be achieved even without fine-tuning.

The compression ratio optimization and the layer-wise optimization can be performed simultaneously in a single pruning framework. For example, the gradient information of a loss function is used to compare the importance of neurons in the whole model (Molchanov et al., 2017). Some methods perform fine-tuning with regularization terms to achieve compression (Liu et al., 2017; Ding et al., 2019; Guo et al., 2021). Although these methods have an advantage in maintaining accuracy while performing pruning, the pruning process itself is time-consuming.

The proposed method in this paper, POEM, is to perform layer-wise optimization. The strength of POEM lies in its strategy for neuron selection and reconstruction based on the output error of the activation function (rather than the error before applying the activation function). It should be noted that Jiang et al. (2018) proposed a pruning method that performs reconstruction based on the activation function outputs, however, their neuron selection is based on heuristic criteria, whereas we purposively select the neurons to minimize the output error based on sound formalization.

## 3    PRUNING WITH OUTPUT ERROR MINIMIZATION

In this section, we introduce the proposed method POEM. POEM is characterized by its neuron (channel) selection and reconstruction strategies based on the output error of the activation function. We define notations in Sec. 3.1, explain the reconstruction strategy of the previous methods (Luo et al., 2017; He et al., 2017; Kamma & Wada, 2021) and that of POEM in Sec. 3.2, our neuron selection strategy in Sec. 3.3, and the algorithm for efficient reconstruction in Sec. 3.4. Although POEM can be applied to both fully connected layers and convolutional layers, we make an explanation using a fully connected layer for simplicity.

### 3.1    PREPARATIONS

Let $n$ be the number of input samples (e.g. images), $a$ and $c$ be the numbers of neurons in the current and the following layers, $\boldsymbol{X}^{\text{ALL}} \in \mathbb{R}^{n \times a}$ be the outputs in the current layer, $\boldsymbol{W}^{\text{ALL}} \in \mathbb{R}^{a \times c}$ be the weight matrix, $f$ be the activation function (e.g. ReLU), and $f'$ be its first derivative. The outputs in the following layer can be obtained by calculating $f(\boldsymbol{Y})$, where $\boldsymbol{Y} = \boldsymbol{X}^{\text{ALL}} \boldsymbol{W}^{\text{ALL}}$. By applying pruning, we remove the columns of $\boldsymbol{X}^{\text{ALL}}$ and the rows of $\boldsymbol{W}^{\text{ALL}}$ corresponding to the pruned neurons, and get $\boldsymbol{X} \in \mathbb{R}^{n \times b}$ and $\boldsymbol{W} \in \mathbb{R}^{b \times c}$, where $b$ denotes the number of preserved neurons.

For a matrix $\boldsymbol{M}$ and an index $i$, $\boldsymbol{M}_{:,i}$ and $\boldsymbol{M}_{i,:}$ denote the $i$-th column and the $i$-th row of $\boldsymbol{M}$, $\boldsymbol{M}_{:,-i}$ denotes a matrix composed of all but the $i$-th columns of $\boldsymbol{M}$, and $\boldsymbol{M}_{-i,:}$ denotes a matrix composed of all but the $i$-th rows of $\boldsymbol{M}$. For a set of indices $\mathbb{A}$, $\boldsymbol{M}_{:,\mathbb{A}}$ and $\boldsymbol{M}_{\mathbb{A},:}$ denote matrices that have the columns and the rows of $\boldsymbol{M}$ corresponding to $\mathbb{A}$. For a vector $\boldsymbol{v}$, $v_i$ denotes its $i$-th element and $v_{\mathbb{A}}$ denotes its elements corresponding to $\mathbb{A}$.

We also define the operators: $\leftarrow$ denotes the assignment, $\setminus$ denotes the set difference, $\circ$ denotes Hadamard product, and $\|\cdot\|_F$ denotes Frobenius norm. For a vector $\boldsymbol{v}$, $\text{diag}(\boldsymbol{v})$ denotes a diagonal matrix having the elements of $\boldsymbol{v}$ on the diagonal entries. For a matrix $\boldsymbol{M}$, $\boldsymbol{M}^2 = \boldsymbol{M} \circ \boldsymbol{M}$.

### 3.2    OUTPUT ERROR MINIMIZATION WITH THE WEIGHTED LEAST SQUARES METHOD

The previous pruning methods perform reconstruction so as to minimize the error of $\boldsymbol{Y}$. This can be formalized as a typical LS problem:

$$\boldsymbol{W}^* = \arg\min_{\boldsymbol{W}} \|\boldsymbol{Y} - \boldsymbol{X}\boldsymbol{W}\|_F^2 . \tag{1}$$

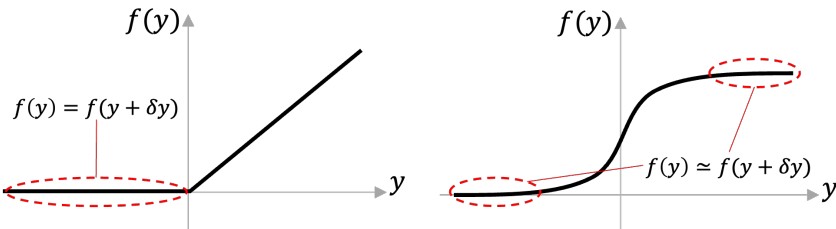

Figure 1: Illustrations of activation functions (left: ReLU, right: Sigmoid). In *flat zones* of these functions, the errors are suppressed. Therefore, it is more important to perform fitting to the elements in non-flat zones.

However, this does not minimize the output error of the activation function. For example, since a ReLU function rounds a negative value to zero, the error of a negative element of $\boldsymbol{Y}$ is canceled by ReLU, unless that element turns positive due to the error. Therefore, it is better to perform reconstruction so as to minimize the error only for the positive elements of $\boldsymbol{Y}$. To generalize, as illustrated in Figure 1, the error of an element is suppressed if that element is in a *flat zone* of the activation function. On the other hand, the error of an element in a *non-flat zone* is not suppressed. For minimizing the output error of the activation function, it is more important to reduce the error for the elements in *non-flat zones*.

This idea can be formalized as a WLS problem. Our reconstruction strategy is to minimize the output error of the activation function:

$$\boldsymbol{W}^* = \arg\min_{\boldsymbol{W}} \|f(\boldsymbol{Y}) - f(\boldsymbol{XW})\|_F^2 . \tag{2}$$

Because it is difficult to solve Eq. (2) directly, we perform relaxation. By performing Taylor expansion for each element of $f(\boldsymbol{Y})$, we get

$$f(\boldsymbol{Y} + \delta\boldsymbol{Y}) = f(\boldsymbol{Y}) + f'(\boldsymbol{Y}) \circ \delta\boldsymbol{Y} + g(\boldsymbol{Y}, \delta\boldsymbol{Y}), \tag{3}$$

where $g(\boldsymbol{Y}, \delta\boldsymbol{Y})$ denotes the second and the higher order terms. The reconstruction error for $\boldsymbol{Y}$ can be written as $\boldsymbol{XW} - \boldsymbol{Y}$. We substitute $\delta\boldsymbol{Y} = \boldsymbol{XW} - \boldsymbol{Y}$ and omit $g$:

$$f(\boldsymbol{XW}) \simeq f(\boldsymbol{Y}) - f'(\boldsymbol{Y}) \circ (\boldsymbol{Y} - \boldsymbol{XW}). \tag{4}$$

By using this approximation, Eq. (2) can be replaced by a WLS problem:

$$\boldsymbol{W}^* = \arg\min_{\boldsymbol{W}} \|f'(\boldsymbol{Y}) \circ (\boldsymbol{Y} - \boldsymbol{XW})\|_F^2 . \tag{5}$$

For each element of $\boldsymbol{Y}$, the error is weighted by the corresponding element of $f'(\boldsymbol{Y})$. Therefore, the error is suppressed for the elements in *flat zones* of $f$.

### 3.3 NEURON SELECTION CRITERIA

When we conduct DNN compression with POEM, we first select and prune the neurons, and then perform reconstruction. We already explained how to perform reconstruction in Sec. 3.2. In this subsection, we explain how to select the neurons to be pruned.

Let $\mathbb{A} \in \{1, \cdots, a\}$ be a subset of column indices of $\boldsymbol{X}^{\mathrm{ALL}}$ corresponding to the preserved neurons. Since we perform reconstruction by solving Eq. (5), we should optimize $\mathbb{A}$ so as to minimize the reconstruction error:

$$\mathbb{A}^* = \arg\min_{\mathbb{A}} \left( \min_{\boldsymbol{W}_{\mathbb{A},:}^{\mathrm{ALL}}} \|f'(\boldsymbol{Y}) \circ \left(\boldsymbol{Y} - \boldsymbol{X}_{:,\mathbb{A}}^{\mathrm{ALL}} \boldsymbol{W}_{\mathbb{A},:}^{\mathrm{ALL}}\right)\|_F^2 \right) \text{ subject to } 1 - \frac{|\mathbb{A}|}{a} \geq p, \tag{6}$$

where $p$ denotes the target compression ratio. Since Eq. (6) is a difficult problem, we solve it in a greedy fashion. Assume that we have already pruned several neurons and got $\boldsymbol{X}$ out of $\boldsymbol{X}^{\mathrm{ALL}}$. Then, we should select the neuron to be pruned next. This is, in other words, to select the $k$-th column of $\boldsymbol{X}$ so as to minimize the error:

$$k^* = \arg\min_{k} \left( \min_{\boldsymbol{W}_{-k,:}} \|f'(\boldsymbol{Y}) \circ (\boldsymbol{Y} - \boldsymbol{X}_{:,-k} \boldsymbol{W}_{-k,:})\|_F^2 \right). \tag{7}$$

---

**Algorithm 1** Memory-efficient algorithm for a WLS problem

---

**Input:** $\boldsymbol{X}$, $\boldsymbol{Y}$, $f'$, number of sampled columns $s$, number of iterations $M$
**Output:** $\boldsymbol{W}$
1: Initialize the weight: $\boldsymbol{W} \leftarrow \arg\min_{\boldsymbol{W}} \|\boldsymbol{Y} - \boldsymbol{X}\boldsymbol{W}\|_F^2$
2: **for** $m = 1, \cdots, M$ **do**
3:    Sample $\mathbb{B}^m$ (the column subset of $\boldsymbol{X}$)   so that   $|\mathbb{B}^m| \leq s$   holds
4:    **for** $j = 1, \cdots, c$ **do**
5:       Perform regression: $\boldsymbol{w}^* \leftarrow \arg\min_{\boldsymbol{w}} \|f'(\boldsymbol{Y}_{:,j}) \circ ((\boldsymbol{Y}_{:,j} - \boldsymbol{X}\boldsymbol{W}_{:,j}) - \boldsymbol{X}_{:,\mathbb{B}^m}\boldsymbol{w})\|^2$
6:       Update the weight: $\boldsymbol{W}_{\mathbb{B}^m,j} \leftarrow \boldsymbol{W}_{\mathbb{B}^m,j} + \boldsymbol{w}^*$
7:    **end for**
8: **end for**

---

In order to solve Eq. (7), we should solve the following problem for all $k$.

$$\boldsymbol{W}^*_{-k,:} = \arg\min_{\boldsymbol{W}_{-k,:}} \|f'(\boldsymbol{Y}) \circ (\boldsymbol{Y} - \boldsymbol{X}_{:,-k}\boldsymbol{W}_{-k,:})\|_F^2 . \tag{8}$$

Since solving Eq. (8) for all $k$ is time-consuming, we once omit $f'(\boldsymbol{Y})$ in order to apply an efficient algorithm (Kamma & Wada, 2021). Then, we have

$$\boldsymbol{W}^*_{-k,:} = \arg\min_{\boldsymbol{W}_{-k,:}} \|\boldsymbol{Y} - \boldsymbol{X}_{:,-k}\boldsymbol{W}_{-k,:}\|_F^2 . \tag{9}$$

Now, we can solve Eq. (7) by using $\boldsymbol{W}^*_{-k,:}$.

$$k^* = \arg\min_{k} \left\|f'(\boldsymbol{Y}) \circ \left(\boldsymbol{Y} - \boldsymbol{X}_{:,-k}\boldsymbol{W}^*_{-k,:}\right)\right\|_F^2 . \tag{10}$$

We prune the $k^*$-th neuron, and continue pruning with the remaining neurons until the target compression ratio is achieved.

## 3.4 An efficient algorithm for the weighted least squares problem

In this subsection, we show how to implement the solution for a WLS problem. We show an algorithm to solve it efficiently, while we explain its details in Appendix B.3.

For a LS problem Eq. (1), we have an one-shot solution:

$$\boldsymbol{W}^* = (\boldsymbol{X}^\top \boldsymbol{X})^{-1}\boldsymbol{X}^\top \boldsymbol{Y}. \tag{11}$$

On the other hand, we do not have such an one-shot solution for a WLS problem. In order to solve Eq. (5), regression should be performed for each column of $\boldsymbol{Y}$ separately:

$$\boldsymbol{W}^*_{:,j} = \arg\min_{\boldsymbol{W}_{:,j}} \|f'(\boldsymbol{Y}_{:,j}) \circ (\boldsymbol{Y}_{:,j} - \boldsymbol{X}\boldsymbol{W}_{:,j})\|^2 . \tag{12}$$

Let $\boldsymbol{G}^j = \mathrm{diag}(f'(\boldsymbol{Y}_{:,j}))$. The solution of Eq. (12) is given by

$$\boldsymbol{W}^*_{:,j} = (\boldsymbol{X}^\top \boldsymbol{G}^{j\,2}\boldsymbol{X})^{-1}\boldsymbol{X}^\top \boldsymbol{G}^{j\,2}\boldsymbol{Y}_{:,j}. \tag{13}$$

We parallelize this for each $j$ to obtain the solution efficiently. Although, implementing Eq. (13) as is would require a large memory space for computing $(\boldsymbol{X}^\top \boldsymbol{G}^{j\,2}\boldsymbol{X})^{-1}$.

Algorithm 1 describes the solution for Eq. (12) with a limited memory space. This algorithm is to approach the solution asymptotically by updating some of the elements of $\boldsymbol{W}$ in turn. To do this, we sample some columns from $\boldsymbol{X}$ and use them for regression. See Appendix B.3 for more details.

## 3.5 Limitation

POEM is supposed to be used for layers with fixed weights, i.e. convolutional layers and fully connected layers. Therefore, layers with dynamic weights, such as (Ramachandran et al., 2019), are not supported.

Table 1: Benchmarking channel selection methods and reconstruction methods. We pruned 32 channels out of 64 channels in "Conv1-1" layer of VGG-16, and compared the MSEs in the following layer. Each cell contains the MSE of the outputs of the activation function (after $f$) or of the values before applying the activation function (before $f$).

| Select.\Rec. | a) No rec. | | b) LS | | c) WLS (ours) | |
|---|---|---|---|---|---|---|
| | after $f$ | before $f$ | after $f$ | before $f$ | after $f$ | before $f$ |
| 1) L1 | 0.2731 | 0.8614 | 0.0379 | 0.0953 | 0.0244 | 0.1750 |
| 2) L2 | 0.2183 | 0.7685 | 0.0141 | 0.0341 | 0.0116 | 0.0660 |
| 3) Lasso | 0.1711 | 0.5537 | 0.0117 | 0.0251 | 0.0095 | 0.0460 |
| 4) GM | 0.2749 | 0.9114 | 0.0550 | 0.1443 | 0.0432 | 0.4143 |
| 5) Error (LS) | 0.2110 | 0.6582 | 0.0067 | 0.0143 | 0.0060 | 0.0215 |
| 6) Error (WLS) | 0.2277 | 0.7226 | 0.0062 | 0.0136 | **0.0055** | 0.0200 |

## 4 EXPERIMENTS

In this section, we show the experiment setups and results. We conducted experiments with VGG-16 (Simonyan & Zisserman, 2015), ResNet-18 (He et al., 2016), and MobileNet (Howard et al., 2017). We mainly used ImageNet (Deng et al., 2009) dataset for evaluation, while a fine-grained classification dataset CUB-200-2011 (Wah et al., 2011) was also used.

### 4.1 DATASET AND AUGMENTATION

ImageNet dataset contains 1.28M training images and 50K validation images (Deng et al., 2009). Since POEM is a data-dependent method, we randomly sampled 5K training images for pruning. For all images, we resized so as to make the shorter side be 256. For training images, we applied $224 \times 224$ random crop and random horizontal flip. For validation images, we applied $224 \times 224$ center crop. CUB-200-2011 dataset contains 6K training images and 5.7K test images. For pruning with POEM, we used all training images. The augmentation settings were the same with those for ImageNet dataset.

### 4.2 PRUNING SETUPS

We performed pruning in convolutional layers of VGG-16, ResNet-18, and MobileNet. Since POEM performs only layer-wise optimization, the compression ratio optimization should be performed externally. We conducted experiments in two scenarios: with and without compression ratio optimization. For configurating compression ratios, we used three methods, AutoML Model Compression (AMC) (He et al., 2018b), Pruning Ratio Optimizer (PRO) (Kamma et al., 2022), and RandomPruning (Li et al., 2022). For AMC, we used the official source code published by the authors of (He et al., 2018b). We implemented the other 2 methods on our own.

### 4.3 TRAINING SETUPS

In the experiments with ImageNet dataset, we fine-tuned VGG-16 and ResNet-18 for 25 epochs, and MobileNet for 150 epochs after pruning, with Stochastic Gradient Descent (SGD) optimizer. For VGG-16, the mini-batch size was set to 128, the learning rate was set to 0.0005, and the weight decay was set to 0.0001. For ResNet-18, the mini-batch size was set to 256, the learning rate was set to 0.001, and the weight decay was set to 0.0001. For MobileNet, the mini-batch size was set to 256, the learning rate was set to 0.0005, and the weight decay was set to 0.00004. For only MobileNet, we applied cosine annealing. The momentum was set to 0.9 for all models.

In the experiments with CUB-200-2011 dataset, we first applied domain adaptation to the pre-trained ResNet-18 model. We initialized the fully connected layer of the ResNet-18 model and conducted training for 100 epochs with SGD optimizer. The learning rate was set to 0.01, and divide it by 10 at 50 epoch. The mini-batch size was set to 64, the weight decay was set to 0.001, and the momentum was set to 0.9. After pruning, we fine-tuned the pruned model using the same settings.

Table 2: The results of pruning with uniform compression ratios. The baseline accuracy of VGG-16 was 0.70272 (top-1) and 0.8946 (top-5) and the accuracy of ResNet-18 was 0.69758 (top-1) and 0.89076 (top-5).

| Model | Method | FLOPs | Acc. w/o fine-tune | | Acc. fine-tuned | |
|---|---|---|---|---|---|---|
| | | | Top-1 | Top-5 | Top-1 | Top-5 |
| VGG-16 | CP-A | -75.8% | 17.33 | 35.042 | 65.754 | 86.868 |
| | REAP-A | -75.8% | 27.926 | 50.688 | 66.642 | 87.466 |
| | POEM-A | -75.8% | **51.492** | **77.472** | **67.268** | **87.68** |
| | CP-B | -75.7% | 39.254 | 70.212 | 67.938 | 88.182 |
| | REAP-B | -75.7% | 52.368 | 77.452 | 68.844 | 88.716 |
| | POEM-B | -75.7% | **59.412** | **82.446** | **69.044** | **88.892** |
| ResNet-18 | CP | -50.4% | 27.136 | 52.894 | 62.0 | 84.182 |
| | REAP | -50.4% | 32.186 | 58.326 | 62.59 | 84.694 |
| | POEM | -50.4% | **42.376** | **69.546** | **62.812** | **84.8** |

## 4.4 LAYER-WISE ANALYSIS

We performed pruning on "Conv1-1" layer of VGG-16, and calculated the MSEs of the outputs of the activation function and the MSEs of the values before applying the activation function. Note that we can decouple and combine the channel selection methods and the reconstruction methods. For example, we could select channels based on L1 norm of the output weights (He et al., 2014) and perform POEM's reconstruction. We used various combinations of channel selection methods and reconstruction methods. For channel selection, we used the ones based on 1) L1 norm of outgoing weights (He et al., 2014), 2) L2 norm of outgoing weights, 3) Lasso regression (He et al., 2017), 4) Geometric Median (GM) of incoming weights (He et al., 2019), 5) LS error (Kamma & Wada, 2021), and 6) WLS error (ours). For reconstruction, we used a) no reconstruction, b) LS (He et al., 2017; Kamma & Wada, 2021), and c) WLS (ours).

Table 1 shows the results. Comparing b) and c), we see that our reconstruction method was better than the previous method in preventing the output error of the activation function. Comparing the row of 6) and the other rows, we can see that our channel selection method also outperformed the other channel selection methods except for the case of not performing reconstruction. These results show the superiority of POEM in both channel selection and reconstruction.

An interesting observation is that the output error of the activation function was the lowest when we used our WLS-based reconstruction, while we suffered relatively large errors of the values before applying the activation function. By performing WLS, we can reduce the errors of the output elements in *non-flat* zones of the activation function, which increases the errors of the elements in *flat zones*. However, as already mentioned, the error in a *flat zone* is suppressed and thus is not harmful.

It is worth noting that the error can be significantly reduced even with the previous reconstruction method, compared to the case of not performing reconstruction. This indicates that reconstruction is crucial for pruning. No matter which channel selection method is chosen, we should perform reconstruction.

## 4.5 PRUNING WITH FIXED COMPRESSION RATIOS

We conducted experiments with VGG-16 and ResNet-18. We performed pruning to compress their convolutional layers with uniform compression ratios, and then conducted fine-tuning. In this section, we compare POEM with two previous methods, CP (He et al., 2017) and REAP (Kamma & Wada, 2021). Both CP and REAP are the pruning methods performing reconstruction but based on the error before applying the activation function. For VGG-16, it is known that the last two convolutional layers are not redundant, and these layers are often excluded from the compression targets (Luo et al., 2017; He et al., 2017). We performed compression in two cases. One is to compress all convolutional layers (Case A), and the other is to compress all but those two layers (Case B).

The results are shown in Table 2. We can see that POEM outperformed the other methods consistently. The performance difference was significant especially before fine-tuning. For example,

Table 3: The results of pruning using AMC. The baseline accuracy of MobileNet was 0.71144 (top-1) and 0.89842 (top-5). The baseline accuracy of VGG-16 was 0.70272 (top-1) and 0.8946 (top-5) and that of ResNet-18 was 0.69758 (top-1) and 0.89076 (top-5).

| Comp. ratio | Model | Method | FLOPs | Acc. w/o fine-tune | | Acc. fine-tuned | |
| | | | | Top-1 | Top-5 | Top-1 | Top-5 |
|---|---|---|---|---|---|---|---|
| AMC | MobileNet | L1&LS | -50.0% | 43.686 | 69.13 | 70.382 | 89.398 |
| | | POEM | -50.0% | **51.984** | **76.648** | **70.488** | **89.492** |
| | | L1&LS | -70.0% | 3.146 | 11.12 | 67.57 | 87.772 |
| | | POEM | -70.0% | **17.476** | **38.74** | **68.104** | **87.992** |
| PRO | VGG-16 | L1&LS | **-75.7%** | 28.356 | 51.396 | 67.134 | 87.804 |
| | | REAP | -75.4% | 57.196 | 80.846 | 69.09 | 88.95 |
| | | POEM | **-75.7%** | **61.956** | **84.18** | **69.484** | **89.02** |
| | ResNet-18 | L1&LS | -50.3% | 42.86 | 70.372 | 64.544 | 86.124 |
| | | REAP | **-50.4%** | 47.648 | 73.964 | 64.874 | **86.33** |
| | | POEM | -50.0% | **52.306** | **77.428** | **65.316** | 86.218 |
| RandomPruning | VGG-16 | L1&LS | **-75.7%** | - | - | 67.438 | 87.854 |
| | | REAP | -75.6% | - | - | 68.78 | 88.614 |
| | | POEM | -74.6% | - | - | **69.296** | **88.92** |
| | ResNet-18 | L1&LS | -50.5% | - | - | 63.606 | 85.386 |
| | | REAP | **-50.7%** | - | - | 63.738 | 85.43 |
| | | POEM | -49.2% | - | - | **64.286** | **85.686** |

in Case A of VGG-16, POEM outperformed REAP by more than 23% margin in top-1 accuracy. Although fine-tuning reduced the accuracy gaps, the model compressed with POEM was still better than the ones compressed with the previous methods.

For VGG-16, the performance gap of POEM and the other methods was more significant in Case A than in Case B. In Case A, we performed pruning on non-redundant layers, while those layers were excluded from compression targets in Case B. This indicates the relative advantage of POEM is greater when we conduct compression for less redundant layers.

## 4.6 PRUNING WITH COMPRESSION RATIO OPTIMIZATION METHODS

We evaluated performance of POEM combined with three compression ratio optimization methods, AMC (He et al., 2018b), PRO (Kamma et al., 2022), and RandomPruning (Li et al., 2022). For AMC, L1 norm-based channel selection and LS-based reconstruction are used as default. We refer this default setting as "L1&LS". We evaluated the pruning performance with L1&LS and with POEM. For PRO and RandomPruning, we used REAP as well.

Table 3 shows the results. With AMC, POEM outperformed L1&LS by 14.3% in top-1 and 27.6% in top-5 accuracy at 70% FLOPs reduction. After fine-tuning, POEM still outperforms L1&LS by 0.5% and 0.2% in top-1 and top-5 accuracy.

With PRO, POEM outperformed the other two methods by a considerable margin before fine-tuning. After fine-tuning, although REAP was better than POEM marginally only in top-5 accuracy of ResNet-18, POEM was the best for the rest.

For RandomPruning, we show only the accuracy after fine-tuning, since fine-tuning is already conducted in the process of compression ratio optimization. For both VGG-16 and ResNet-18, POEM outperformed the second best method REAP by the margin of 0.5% and 0.2% in top-1 and top-5 accuracy.

The difference of POEM and L1&LS for MobileNet was larger at higher compression ratio. After a significant amount of fine-tuning (150 epochs), we can still see clear accuracy gap. This indicates that the performance superiority of POEM becomes more significant at higher compression ratio.

Another remarkable observation is that the pruning performance was poor when we used L1&LS and PRO. The result for VGG-16 was worse than even some results with the fixed compression ratios. This is most likely because PRO is a type of method that narrows down the search space

Table 4: The results with CUB-200-2011 dataset. The baseline accuracy of ResNet-18 was 0.7571 (top-1) and 0.9283 (top-5).

| Model | Method | FLOPs | Acc. w/o fine-tune | | Acc. fine-tuned | |
|-------|--------|-------|--------|--------|--------|--------|
| | | | Top-1 | Top-5 | Top-1 | Top-5 |
| ResNet-18 | CP | -50.4% | 62.35 | 85.57 | 70.84 | 90.24 |
| | REAP | -50.4% | 64.70 | 86.71 | 71.91 | 90.54 |
| | POEM | -50.4% | **69.58** | **89.86** | **72.33** | **90.81** |
| | Scratch | -50.4% | - | - | 62.94 | 84.84 |
| | CP | -75.3% | 20.27 | 43.66 | 65.82 | 88.09 |
| | REAP | -75.3% | 21.67 | 45.65 | 66.79 | 88.64 |
| | POEM | -75.3% | **47.91** | **76.38** | **68.08** | **89.21** |
| | Scratch | -75.3% | - | - | 58.14 | 83.12 |

of compression ratios sequentially. Once the model accuracy is severely damaged, it is no more possible to find a good solution. On the other hand, we could find much better solutions for both VGG-16 and ResNet-18 when we used POEM and PRO. Thus, combining a compression ratio optimization method with a better layer-wise optimization method results in more effective pruning.

### 4.7 EXPERIMENTS WITH A FINE-GRAINED CLASSIFICATION DATASET

We conducted experiments with ResNet-18 model and CUB-200-2011 dataset. We first fine-tuned the pre-trained model using CUB-200-2011 dataset, then performed pruning with the uniform compression ratios in all convolutional layers. We also evaluated the performance of the models with the pruned architectures but trained from scratch. For the "trained-from-scratch" models, we conducted training for 400 epochs. The learning rate was set to 0.1, and was divided by 10 at 200 and 300 epochs.

The results are shown in Table. 4. We can see that POEM was better than the other methods consistently. At 75% FLOPs reduction ratio, POEM outperformed REAP by 26.2% and by 30.7% in top-1 and top-5 accuracy, respectively. After fine-tuning, POEM still outperformed REAP by approximately 1.2% and 0.5% in top-1 and top-5 accuracy.

It is worth noting that the performance of trained-from-scratch models were much worse than the pruned and fine-tuned models. In the case of training them from scratch, it is difficult to generalize for such a small dataset. These results let us rethink the value of pruning. Nowadays, it is believed that the value of pruning lies in the search for efficient architecture rather than the weight parameters obtained by pruning (and fine-tuning), based on the research results in (Liu et al., 2019). Although this is true for the large scale datasets, such as ImageNet, CIFAR-10 (Krizhevsky et al.), and so on, it is not the case with small scale datasets with which training from scratch easily falls into bad local minima. In many real scenarios, the model needs to be trained with limited amount of data due to poor availability of the data, annotation cost, and other reasons. In such cases, it is an effective approach to use the pruned and fine-tuned models, rather than the trained-from-scratch models.

## 5 CONCLUSION

In this paper, we presented Pruning with Output Error Minimization (POEM), the method to conduct pruning and perform reconstruction to minimize the output error of the activation function rather than the error of the value before applying the activation function. In experiments using well-known DNN models for image recognition, VGG-16, ResNet-18, and MobileNet, we confirmed that the proposed method can perform compression with smaller errors than the previous methods. We also confirmed that the proposed method, combined with the compression ratio optimization methods, enables more effective compression. In Appendices, we show additional information for experiment reproducibility and implementation.

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

# A    ADDITIONAL INFORMATION FOR EXPERIMENT REPRODUCIBILITY

## A.1    WHERE WE GOT THE PRE-TRAINED MODELS

In the experiments, we used off-the-shelf models. The trained weights for VGG-16, ResNet-18, and MobileNet can be downloaded via the following URLs.

- `http://www.robots.ox.ac.uk/~vgg/software/very_deep/caffe/VGG_ILSVRC_16_layers.caffemodel`
- `https://download.pytorch.org/models/resnet18-5c106cde.pth`
- `https://hanlab.mit.edu/projects/amc/external/mobilenet_imagenet.pth.tar`

## A.2    PRUNING FOR BRANCHED PATHS OF RESNET

For the layers having skip connections of the ResNet model, we cannot perform pruning as is, because the number of channels should be the same in the both ends of skip connections. We have several ways to avoid this problem. The first is to perform pruning only for the layers without skip connections. The second is to add a layer that only samples the channels to be preserved (He et al., 2017), as shown in Fig. 2. The third is to conduct pruning at the both ends of the skip connection (Luo & Wu, 2020). We took the the second one.

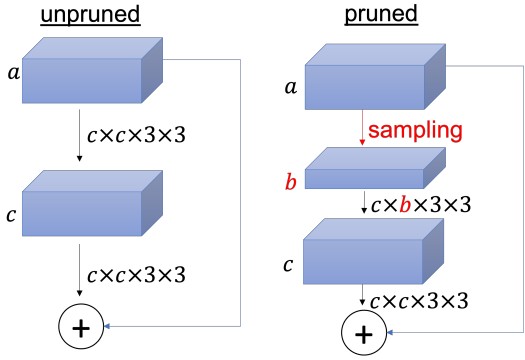

Figure 2:  Instead of removing the channels in the fist layer, we sample $b$ channels to be preserved.

# B    ADDITIONAL INFORMATION ON IMPLEMENTATION OF POEM

## B.1    HOW TO EXTEND TO CONVOLUTIONAL LAYERS

In most major DNN frameworks, such as Pytorch, a feature map in the convolutional layer is expanded to be a matrix by using "im2col" function (Paszke et al., 2019). With the expanded matrices, the convolutional operation can be replaced by a simple dot product, such that $Y = XW$. Therefore, POEM can be applied to the convolutional layers in the similar with to the fully connected layers. The only difference is that with the expanded feature map, each channel corresponds to several columns of the matrix. Thus, for convolutional layers, we need to remove several columns of $X$ at each time.

## B.2    PRUNING FOR WHOLE MODEL

Once we conduct pruning on a layer, the outputs in the subsequent layers are more or less affected. Assume that we have conducted pruning in a previous layer and are now pruning in the current layer. The original outputs in the current layer, $X$, has become $X' \neq X$. We use the new outputs $X'$, whereas we still use the original $Y$ for neuron selection and reconstruction so that the original performance of the model can be the best preserved.

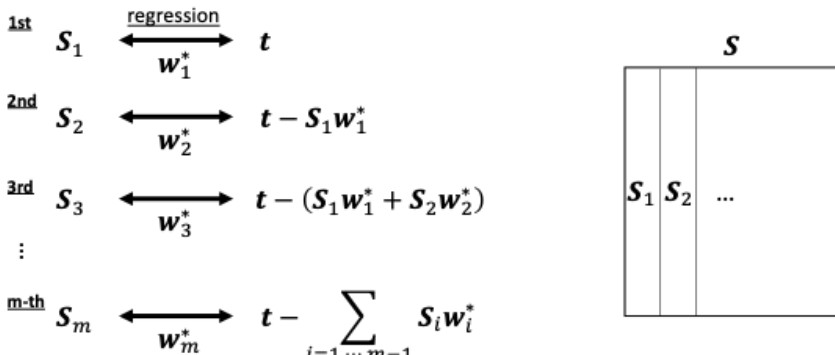

Figure 3: The concept of Algorithm 1.

### B.3   DETAILS OF ALGORITHM 1

Algorithm 1 aims to solve Eq. (12) with a limited memory space. Let us simplify the notations: $\boldsymbol{S} = \boldsymbol{G}^j \boldsymbol{X}$, $\boldsymbol{t} = \boldsymbol{G}^j \boldsymbol{Y}_{:,j}$, and $\boldsymbol{u} = W_{:,j}$. Then, Eq. (12) can be rewritten as a typical LS problem:

$$\boldsymbol{u}^* = \arg\min_{\boldsymbol{u}} \|\boldsymbol{t} - \boldsymbol{S}\boldsymbol{u}\|^2. \tag{14}$$

The solution is given by

$$\boldsymbol{u}^* = \left(\boldsymbol{S}^\top \boldsymbol{S}\right)^{-1} \boldsymbol{S}^\top \boldsymbol{t}. \tag{15}$$

However, Eq. (15) would require a large memory space for computing $\left(S^\top S\right)^{-1}$.

To solve Eq. (14) with a limited memory space, we sample a small number of columns from $\boldsymbol{S}$ to perform regression, as shown in Fig. 3. Let $\boldsymbol{S}_1, \boldsymbol{S}_2, \cdots$ be the matrices having a few columns sampled from $\boldsymbol{S}$. We first perform regression with $\boldsymbol{S}_1$ and $\boldsymbol{t}$:

$$\boldsymbol{w}_1^* = \arg\min_{\boldsymbol{w}_1} \|\boldsymbol{t} - \boldsymbol{S}_1 \boldsymbol{w}_1\|^2. \tag{16}$$

The error for $\boldsymbol{t}$ is given by $\boldsymbol{t} - \boldsymbol{S}_1 \boldsymbol{w}_1^*$. Next, we perform regression so as to fill this error using $\boldsymbol{S}_2$:

$$\boldsymbol{w}_2^* = \arg\min_{\boldsymbol{w}_2} \|(\boldsymbol{t} - \boldsymbol{S}_1 \boldsymbol{w}_1^*) - \boldsymbol{S}_2 \boldsymbol{w}_2\|^2. \tag{17}$$

In this way, we continue to perform regression so as to fill the error for $\boldsymbol{t}$:

$$\boldsymbol{w}_m^* = \arg\min_{\boldsymbol{w}_m} \left\|\left(\boldsymbol{t} - \sum_{i=1,\cdots,m-1} \boldsymbol{S}_i \boldsymbol{w}_i^*\right) - \boldsymbol{S}_m \boldsymbol{w}_m\right\|^2. \tag{18}$$

Once we finish all of the $\boldsymbol{S}$-s, we continue from $\boldsymbol{S}_1$ again. After enough cycles, the error for $\boldsymbol{t}$ will be minimized. The approximate solution for Eq. (14) can be calculated from the $\boldsymbol{w}^*$-s.

This algorithm is memory-efficient. The solution of Eq. (18) is given by

$$\boldsymbol{w}_m^* = \left(\boldsymbol{S}_m^\top \boldsymbol{S}_m\right)^{-1} \boldsymbol{S}_m^\top \left(\boldsymbol{t} - \sum_{i=1,\cdots,m-1} \boldsymbol{S}_i \boldsymbol{w}_i^*\right). \tag{19}$$

Since $\left(\boldsymbol{S}_m^\top \boldsymbol{S}_m\right)^{-1}$ is a small matrix, calculating it does not require large memory. Therefore, this algorithm is suitable for parallel calculation.

In Algorithm 1, we have two hyper parameters: $s$ is the number of sampled columns, and $M$ is how many times we perform column sampling and regression. In our experiments, we set $s = 9$, and $M = 5b$, where $b$ is the number of preserved channels.

Table 5: Benchmarking channel selection methods and reconstruction methods. We pruned 50% of channels in "Conv2-1", "Conv3-1", "Conv4-1" layers of VGG-16, and compared the MSEs in the following layer. Each cell contains the output error of $f(\boldsymbol{Y})$, and the value in parenthesis is the error of $\boldsymbol{Y}$ (the error before applying the activation function).

| | Conv2-1 | | | | | |
|---|---|---|---|---|---|---|
| Select.\Rec. | a) No rec. | | b) LS | | c) WLS (ours) | |
| | after $f$ | before $f$ | after $f$ | before $f$ | after $f$ | before $f$ |
| 1) L1 | 1.2695 | 5.8902 | 0.2700 | 0.8461 | 0.2570 | 1.0567 |
| 2) L2 | 1.2433 | 5.9156 | 0.2651 | 0.8305 | 0.2516 | 1.0331 |
| 3) Lasso | 1.3555 | 4.8980 | 0.2670 | 0.8175 | 0.2519 | 1.0692 |
| 4) GM | 1.2001 | 5.7369 | 0.2538 | 0.7958 | 0.2422 | 0.9789 |
| 5) Error (LS) | 1.2060 | 5.2416 | 0.2242 | 0.7000 | 0.2143 | 0.8743 |
| 6) Error (WLS) | 1.1688 | 5.4151 | 0.2168 | 0.6827 | **0.2048** | 0.8634 |
| | Conv3-1 | | | | | |
| Select.\Rec. | a) No rec. | | b) LS | | c) WLS (ours) | |
| | after $f$ | before $f$ | after $f$ | before $f$ | after $f$ | before $f$ |
| 1) L1 | 4.0288 | 21.332 | 1.0621 | 3.6876 | 0.9978 | 4.5270 |
| 2) L2 | 4.0986 | 21.388 | 1.0588 | 3.6779 | 0.9959 | 4.5660 |
| 3) Lasso | 4.5589 | 18.946 | 1.1959 | 4.1310 | 1.0997 | 5.1430 |
| 4) GM | 4.1844 | 21.933 | 1.0842 | 3.7897 | 1.0179 | 4.7850 |
| 5) Error (LS) | 3.9163 | 21.916 | 0.9818 | 3.4051 | 0.9260 | 4.1682 |
| 6) Error (WLS) | 3.7945 | 21.145 | 0.9688 | 3.3444 | **0.9094** | 4.1231 |
| | Conv4-1 | | | | | |
| Select.\Rec. | a) No rec. | | b) LS | | c) WLS (ours) | |
| | after $f$ | before $f$ | after $f$ | before $f$ | after $f$ | before $f$ |
| 1) L1 | 3.3865 | 38.488 | 1.2485 | 7.6413 | 1.1966 | 13.866 |
| 2) L2 | 3.4854 | 40.214 | 1.2656 | 7.7990 | 1.2154 | 14.327 |
| 3) Lasso | 3.5152 | 34.330 | 1.3719 | 8.2762 | 1.3325 | 14.780 |
| 4) GM | 3.3631 | 43.098 | 1.2504 | 7.7296 | 1.1997 | 13.040 |
| 5) Error (LS) | 2.9017 | 35.803 | 1.1540 | 6.8973 | 1.0948 | 11.518 |
| 6) Error (WLS) | 2.9430 | 36.686 | 1.1132 | 6.7026 | **1.0527** | 10.885 |

## B.4 AN EFFICIENT ALGORITHM FOR SOLVING EQ. (9)

We implemented the algorithm proposed in (Kamma & Wada, 2021) to solve Eq. (9) for all $k$ efficiently. Here, we define additional notations: $\mathrm{norm}(\boldsymbol{v})$ denotes the L2 norm operator for a vector $\boldsymbol{v}$, and $\boldsymbol{M}^+$ denotes a Moore-Penrose pseudo inverse of a matrix $\boldsymbol{M}$.

We assume $\boldsymbol{X} = \boldsymbol{X}^{\mathrm{ALL}}$ and $\boldsymbol{W} = \boldsymbol{W}^{\mathrm{ALL}}$ for simplicity. The motivation for solving Eq. (9) is to obtain the following $\boldsymbol{Z}^k$ for all $k$.

$$\boldsymbol{Z}^k = \boldsymbol{Y} - \boldsymbol{X}_{:,-k}\boldsymbol{W}^*_{-k,:}. \tag{20}$$

We can compute $\boldsymbol{Z}^k$ directly without actually computing $\boldsymbol{W}^*_{-k,:}$. To do so, we first compute the following.

$$\bar{\boldsymbol{X}} = (\boldsymbol{X}^+)^\top. \tag{21}$$

Let $\boldsymbol{D} = \mathrm{diag}(\boldsymbol{d})$, where $d_i = \mathrm{norm}(\bar{\boldsymbol{X}}_{:,i})$. We calculate the following.

$$\boldsymbol{R} = \bar{\boldsymbol{X}}\boldsymbol{D}^{-2}. \tag{22}$$

Then, $\boldsymbol{Z}^k$ can be computed as

$$\boldsymbol{Z}^k = \boldsymbol{R}_{:,k}\boldsymbol{W}_{k,:}. \tag{23}$$

Table 6: The MSEs in Conv2-1, Conv3-1, Conv4-1 after pruning 50% of the chennels in Conv1-1.

| | Conv2-1 | | | | | |
|---|---|---|---|---|---|---|
| Select.\Rec. | a) No rec. | | b) LS | | c) WLS (ours) | |
| | after $f$ | before $f$ | after $f$ | before $f$ | after $f$ | before $f$ |
| 1) L1 | 0.7564 | 2.4049 | 0.1624 | 0.4946 | 0.1041 | 0.3181 |
| 2) L2 | 0.7395 | 2.4725 | 0.0593 | 0.1789 | 0.0444 | 0.1350 |
| 3) Lasso | 0.6858 | 2.2340 | 0.0586 | 0.1959 | 0.0382 | 0.1201 |
| 4) GM | 0.7189 | 2.2748 | 0.2121 | 0.6496 | 0.1555 | 0.4798 |
| 5) Error (LS) | 0.7177 | 2.4643 | 0.0374 | 0.1167 | 0.0293 | 0.0904 |
| 6) Error (WLS) | 0.8078 | 2.9315 | 0.0338 | 0.1022 | **0.0279** | 0.0845 |

| | Conv3-1 | | | | | |
|---|---|---|---|---|---|---|
| Select.\Rec. | a) No rec. | | b) LS | | c) WLS (ours) | |
| | after $f$ | before $f$ | after $f$ | before $f$ | after $f$ | before $f$ |
| 1) L1 | 1.9539 | 8.0325 | 0.3933 | 1.4787 | 0.2551 | 0.8983 |
| 2) L2 | 1.9751 | 8.3912 | 0.1518 | 0.5384 | 0.1141 | 0.3973 |
| 3) Lasso | 1.9891 | 8.2181 | 0.1649 | 0.6534 | 0.1071 | 0.3892 |
| 4) GM | 1.8400 | 8.0139 | 0.5241 | 1.9235 | 0.3897 | 1.3814 |
| 5) Error (LS) | 1.9232 | 8.5490 | 0.0987 | 0.3580 | 0.0777 | 0.2733 |
| 6) Error (WLS) | 2.1078 | 10.0640 | 0.0850 | 0.2967 | **0.0704** | 0.2430 |

| | Conv4-1 | | | | | |
|---|---|---|---|---|---|---|
| Select.\Rec. | a) No rec. | | b) LS | | c) WLS (ours) | |
| | after $f$ | before $f$ | after $f$ | before $f$ | after $f$ | before $f$ |
| 1) L1 | 1.5469 | 10.8463 | 0.2801 | 1.8480 | 0.1752 | 1.1079 |
| 2) L2 | 1.4809 | 11.1189 | 0.1097 | 0.6932 | 0.0811 | 0.5015 |
| 3) Lasso | 1.5120 | 11.8430 | 0.1208 | 0.8532 | 0.0790 | 0.5054 |
| 4) GM | 1.4223 | 10.7641 | 0.3933 | 2.5382 | 0.2857 | 1.7862 |
| 5) Error (LS) | 1.3722 | 11.2451 | 0.0680 | 0.4377 | 0.0540 | 0.3383 |
| 6) Error (WLS) | 1.5128 | 15.2813 | 0.0564 | 0.3459 | **0.0468** | 0.2847 |

## C    EXTRA RESULTS

### C.1    EXTRA LAYER-WISE ANALYSIS

We conducted layer-wise analyses on several layers other than Conv1-1 layer. Table 5 shows the results. Similarly with Table 1, our WLS-based reconstruction and neuron selection was better than the other approaches.

### C.2    IMPACT OF PRUNING IN A LAYER ON THE SUBSEQUENT LAYERS

We pruned 50% of the channels in Conv1-1 (the first convolutional layer) of VGG-16, and calculated the MSEs in several subsequent layers (Conv2-1, Conv3-1, Conv4-1). Table 6 shows the results. We can see that the proposed channel selection method and the reconstruction method are better than the others. This trend is consistent with Table 5. The trend is that the smaller the error in Conv1-1 (where pruning was performed) was, the smaller the errors in the subsequent layers were.

### C.3    IMPORTANCE OF CHANNEL SELECTION

We performed pruning in all convolutional layers of VGG-16 to benchmark the channel selection criteria. For channel selection, we used all 6 criteria that are listed in Table 1 (L1, L2, Lasso, GM, Error (LS), Error (WLS)); we combined these channel selection options with our WLS-based reconstruction. The compression ratios were set to 50% in all target layers.

Table 7 shows the results. The accuracy after pruning without fine-tuning is reported. With our channel selection criteria (Error (WLS)), the accuracy was the best preserved. These results show the importance of our channel selection criteria as well as our reconstruction.

Table 7: The comparison of channel selection criteria combined with our WLS-based reconstruction. We conducted pruning with fixed compression ratios (50% in one case, 60% in another case) in all layers. The accuracy before fine-tuning is reported in this table. The baseline accuracy of VGG-16 was 0.70272 (top-1) and 0.8946 (top-5).

| Model | Select criteria | FLOPs | w/o fine-tune | |
| --- | --- | --- | --- | --- |
| | | | Top-1 | Top-5 |
| VGG-16 | L1 | -73.5% | 41.31 | 68.85 |
| | L2 | -73.5% | 45.62 | 72.62 |
| | Lasso | -73.5% | 43.35 | 70.82 |
| | GM | -73.5% | 48.38 | 74.75 |
| | Error (LS) | -73.5% | 50.19 | 76.38 |
| | Error (WLS) | -73.5% | **51.20** | **77.04** |
| | L1 | -82.6% | 23.68 | 48.24 |
| | L2 | -82.6% | 29.05 | 55.92 |
| | Lasso | -82.6% | 27.03 | 52.63 |
| | GM | -82.6% | 25.36 | 50.77 |
| | Error (LS) | -82.6% | 37.63 | 64.77 |
| | Error (WLS) | -82.6% | **39.13** | **66.82** |

Table 8: The results of pruning with uniform compression ratios. The accuracy after 100 epochs of fine-tuning is reported in this table. The baseline accuracy of VGG-16 was 0.70272 (top-1) and 0.8946 (top-5) and the accuracy of ResNet-18 was 0.69758 (top-1) and 0.89076 (top-5).

| Model | Method | FLOPs | Acc. fine-tuned | |
| --- | --- | --- | --- | --- |
| | | | Top-1 | Top-5 |
| VGG-16 | CP-A | -75.8% | 68.89 | 89.08 |
| | REAP-A | -75.8% | 0.6945 | 0.8928 |
| | POEM-A | -75.8% | **0.6998** | **0.8941** |
| | CP-B | -75.7% | 70.47 | 89.59 |
| | REAP-B | -75.7% | 70.68 | 89.85 |
| | POEM-B | -75.7% | **71.01** | **89.96** |
| ResNet-18 | CP | -50.4% | 64.15 | 85.66 |
| | REAP | -50.4% | 64.46 | 85.89 |
| | POEM | -50.4% | **64.77** | **85.97** |

## C.4 EXTRA FINE-TUNING ON VGG-16 AND RESNET-18

We conducted longer fine-tuning (100 epochs) for the pruned VGG-16 and ResNet-18 models. The results are shown in Table 8 (with uniform compression ratios) and Table 9 (with compression ratio optimization). Except for the top-5 error of ResNet-18 with RandomPruning, POEM outperformed the previous methods. These results show the trend: the higher the accuracy was preserved during the pruning procedure, the higher the accuracy becomes after fine-tuning.

## C.5 RESULTS WITH RESNET-50 (WITH FIXED PRUNING RATIOS)

We evaluated POEM with a larger model ResNet-50. We conducted pruning in 32 layers that do not have skip connections. The pruning ratios were set uniformly over all target layers and were tuned so as to make the FLOPs reduction ratio approximately 50%.

The results are shown in Table 10. Before fine-tuning, POEM outperformed the second best method REAP significantly: the top-1 accuracy was 59.75% (POEM) and 44.60% (REAP). After fine-tuning, POEM still outperformed REAP in top-1 accuracy, although the top-5 accuracy of CP was slightly better than POEM slightly. To confirm the winner for ResNet-50, more experiments are necessary with higher pruning ratios, and with pruning ratio optimizations.

Table 9: The results of pruning with pruning ratio optimization. The accuracy after 100 epochs of fine-tuning is reported in this table. The baseline accuracy of VGG-16 was 0.70272 (top-1) and 0.8946 (top-5) and that of ResNet-18 was 0.69758 (top-1) and 0.89076 (top-5).

| Comp. ratio | Model | Method | FLOPs | Acc. fine-tuned | |
| --- | --- | --- | --- | --- | --- |
| | | | | Top-1 | Top-5 |
| PRO | VGG-16 | L1&LS | **-75.7**% | 70.01 | 89.46 |
| | | REAP | -75.4% | 70.95 | 89.97 |
| | | POEM | **-75.7**% | **71.19** | **90.00** |
| | ResNet-18 | L1&LS | -50.3% | 66.00 | 86.89 |
| | | REAP | **-50.4**% | 65.81 | 86.90 |
| | | POEM | -50.0% | **66.27** | **87.08** |
| RandomPruning | VGG-16 | L1&LS | **-75.7**% | 70.00 | 89.37 |
| | | REAP | -75.6% | 70.58 | 89.73 |
| | | POEM | -74.6% | **70.99** | **89.94** |
| | ResNet-18 | L1&LS | -50.5% | 65.44 | **86.77** |
| | | REAP | **-50.7**% | 65.47 | 86.51 |
| | | POEM | -49.2% | **65.68** | 86.62 |

Table 10: The results of pruning for ResNet-50 with uniform compression ratios. The baseline accuracy of ResNet-50 was 76.12 (top-1) and 92.86 (top-5).

| Model | Method | FLOPs | Acc. w/o fine-tune | | Acc. fine-tuned | |
| --- | --- | --- | --- | --- | --- | --- |
| | | | Top-1 | Top-5 | Top-1 | Top-5 |
| ResNet-50 | CP | -49.9% | 42.74 | 68.55 | 73.31 | **91.61** |
| | REAP | -49.9% | 44.60 | 70.55 | 73.50 | 91.42 |
| | POEM | -49.9% | **59.75** | **83.68** | **73.73** | 91.58 |

