# OpenReview forum: "Pruning with Output Error Minimization for Producing Efficient Neural Networks"
_ICLR.cc/2023/Conference — Submitted to ICLR 2023_

### Official Review · Reviewer_fHTY · 2022-10-23

**Confidence:** 4
**Correctness:** 3
**Technical Novelty And Significance:** 2
**Empirical Novelty And Significance:** 2
**Recommendation:** 5

**Clarity, Quality, Novelty And Reproducibility:**

- For post-training quantization, similar approaches have already been introduced. For example, BRECQ and QDrop employ block-level objective function that can include not only activation function but also even neighboring linear layers. Then, an optimizer is employed (instead of analytical solution introduced in this manuscript) to optimize the objective function. Can structured pruning also follow such an approach (i.e., block-level reconstruction based on an optimizer)?
- Why is the proposed method useful for structured pruning? What about applying POEM principles to quantization or fine-grained pruning? What is the motivation to choose structured pruning while POEM itself can be applied to different model compression schemes?
- Can the authors provide differences and advantages compared to recently published structured pruning techniques?

**Strength And Weaknesses:**

*Strength
- The need to include activation function is clear (especially as shown in Figure 1)
- Simplified reconstruction strategy is intuitive and computationally reasonable.
- WLS can be combined with various pruning criteria and pruning ratio decision methods.

*Weakness
- Even though model accuracy without fine-tuning is impressive by POEM, after fine-tuning, the impact of POEM seems to be marginal.
- Experiments are limited to old models. What about the chances to apply POEM to Transformers? Is the method specific to vision tasks?
- Limitations of the proposed techniques are not clearly described.
- There are numerous structured pruning methods recently published. Comparison with those methods are not provided (such as SNIP?)

**Summary Of The Paper:**

The authors propose Pruning with Output Error Minimization (POEM) technique to improve model accuracy after structured pruning. In order to include activation function in the objective function to be optimized, the authors suggest the weighted least squares method while the authors claim the previous methods minimize the error of the value before applying the activation function. One example is ReLU function when negative inputs are converted into zeros such that such highly non-linear functions need to be included during selecting neurons to be pruned. Throughout various CNN models, POEM significantly outperforms the previous methods when fune-tuning is not employed. Even after fine-tuning, POEM usually yields better top-1 and top-5 accuracy.

**Summary Of The Review:**

POEM idea itself is quite interesting and intuitive while this reviewer believes that the underlying principles (that can include activation functions in the objective function) have already been considered in other model compression techniques (especially for post-training quantization, e.g., BRECQ). Also, this reviewer is wondering why structured pruning has been investigated in the manuscript while POEM can be extended to other model compression ideas (experimental results are not very impressive after fine-tuning).

---

> ### Author Response · Authors · 2022-11-18
> **Difference from suggested quantization methods.**
>
> We appreciate a lot for the review. Our rebuttals are as follows. We hope the rebuttal helps to solve the concerns and to increase the score.
>
> ## Accuracy gap after fine-tuning
> The accuracy gap is indeed less significant after fine tuning than before fine-tuning. However, in most of the experiments, we outperform the previous methods by some margin (e.g. 0.5% gap for MobileNet, at -70% FLOPs). We believe that these results are sufficient for claiming the superiority of POEM.
>
> ## Experiments for newer models (e.g. Transformer)
> We added experiments with ResNet-50 (Appendix C.5). For even larger models and other types of architecture like Transformers, we could not conduct experiments due to our time and resource limitation.
>
> ## Comparison to other pruning method (SNIP, etc)
> We wanted to make benchmark for layer-wise methods, thus we chose CP and REAP for comparison. We believe our results are meaningful enough. Although we there are many other pruning methods, we found that fair comparison is very difficult unless we replicate them by ourselves, due to difference in model references, hyper parameters, and so on. In addition, the experimental details are often not clear in many cases. Therefore, we wanted to evaluate the previous methods by ourselves. For now, the results in the updated paper are the best we can show.
>
> ## Difference from BRECQ and QDrop
> BRECQ [4] and QDrop [5] aim to minimize the loss function directly, and in this term their reconstruction criteria is purposive. On the other hand, POEM's criteria is on layer-wise MSE. Therefore, the underlying ideas are different between POEM and these 2 methods. However, we cited BRECQ and QDrop in the updated paper because they are definitely related in terms of DNN compression.
>
> ## Can structured pruning also follow block-level reconstruction like BRECQ and QDrop
> As we mentioned in the reply to z5BU, POEM can be extended so as to minimize the final output error directly, while we focus on minimizing the layer-wise error for now. Therefore, it should be possible to extend POEM (and some other structured pruning methods) to perform block-level reconstruction, which is in between the layer-wise reconstruction and the final output error-based reconstruction.
>
> ## Why limited to structured pruning?
> The principles of POEM can be applied to other approaches, such as unstructured pruning, quantization, and so on. The main motivation to chose structured pruning is that the compressed model can be deployed without special hardware/libraries. We added the statements of this motivation in "Related works" section in the updated paper.
>
> ### References
> [3] Lee et al. SNIP: SINGLE-SHOT NETWORK PRUNING BASED ON CONNECTION SENSITIVITY. ICLR2019.
> [4] Li et al. BRECQ: Pushing the Limit of Post-Training Quantization by Block Reconstruction. ICLR2021.
> [5] Wei et al. QDrop: Randomly Dropping Quantization for Extremely Low-bit Post-Training Quantization. ICLR2022.

---

### Official Review · Reviewer_9TdV · 2022-10-24

**Confidence:** 3
**Correctness:** 3
**Technical Novelty And Significance:** 2
**Empirical Novelty And Significance:** 2
**Recommendation:** 5

**Clarity, Quality, Novelty And Reproducibility:**

- I think this paper should be properly compared with [Jiang, IJCAI’18] as described above.
- I think additional experiments are necessary to show the scalability and effectiveness of the method (see above).
- The method is introduced mainly for fully connected layers, but it would be better to (briefly) describe its extension to convolutional layers.

**Strength And Weaknesses:**

Strength
- The idea that considers the reconstruction errors of post-activation values is simple yet well-motivated.
- The authors validate the proposed layer-wise optimization under various experimental setups (e.g., at fixed compression ratios, combined with recent compression ratio optimization methods). Furthermore, the results before as well as after finetuning are presented to show the effectiveness of the method.
- The paper is easy to follow and fairly clearly written. I think most members of the community would be able to easily dissect it without requiring substantial prior knowledge.

Weaknesses
- I am not sure whether good reconstruction of the features leads to (or is actually related to) good final accuracy after finetuning. One may think that because finetuning significantly recovers the performance, how well the features are reconstructed may have less impact. I would recommend the authors to show lower MSEs in Table 1 result in better accuracies after finetuning. Furthermore, I was wondering if the result trend in Table 1 could be also observable in higher layers and/or other networks.
- I am not sure whether omitting f’(Y) when deriving (9) from (8) is okay. I think a key element of the proposed method is applying the derivative of nonlinearity to a typical LS problem of (1), changing it into the weighted LS of (5). However, the omission of f’(Y) in (9) looks similar to the typical LS setup of (1) and may be unreasonable according to the authors’ claim in Section 3.1. Could the authors provide why this approximation is valid?
- The main idea of this paper looks very closer to that of [Jiang, IJCAI’18]: “Taking the widely used activation function ReLU r(x) = max(0,x) as an example, the Euclidean distance can be large between two negative input values, but would become 0 after activation. Thus, nonlinear reconstruction error (NRE), which computes the Euclidean distance between nonlinear activation values of the unpruned model and those of the pruned model, can be a more reasonable metric than LRE when performing layer-wise pruning.” What are the differences and merits of the proposed approach over [Jiang, IJCAI’18]?
    - Jiang et al., Efficient DNN Neuron Pruning by Minimizing Layer-wise Nonlinear Reconstruction Error, IJCAI'18
- I think the experiments are conducted on relatively shallow networks (i.e., VGG-16, ResNet-18). Recently, many pruning algorithms have been tested on ResNet-50 and/or ResNet-101 [Luo et al., ICCV’17; Liu et al., ICCV’19; He et al, CVPR’19; Li et al., CVPR’22]. I would recommend the authors to present additional results with deeper depth (at least, ResNet-50) to demonstrate the scalability and effectiveness of the proposed method.
    - Luo et al., ThiNet: A Filter Level Pruning Method for Deep Neural Network Compression, ICCV’17
    - Liu et al., MetaPruning: Meta Learning for Automatic Neural Network Channel Pruning, ICCV'19
    - He et al., Filter Pruning via Geometric Median for Deep Convolutional Neural Networks Acceleration, CVPR’19
    - Li et al., Revisiting Random Channel Pruning for Neural Network Compression, CVPR’22


**Summary Of The Paper:**

This paper introduces a layer-wise pruning method by minimizing the reconstruction errors of nonlinear outputs. Unlike the previous methods that compute the errors before the nonlinear function, the proposed method does it after the nonlinearity and focuses on reducing the errors in the non-saturating regions. The authors formulate their idea as a weighted least squares method and also introduce an efficient way to solve it. The proposed approach is validated with relatively shallow networks.

**Summary Of The Review:**

I think the main idea is simple and intuitive, but the empirical results are quite weak and some claims should be better supported. I thus find it difficult to argue for acceptance of the work.

---

> ### Author Response · Authors · 2022-11-18
> **Difference from [Jiang, IJCAI’18]; Additional experiments.**
>
> ## I would recommend the authors to show lower MSEs in Table 1 result in better accuracies after finetuning.
> We already show such results (although partially) in the paper. POEM uses "(6) WLS error"-based channel selection" and "(c) WLS"-based reconstruction in Table 1. The previous method CP uses "(3) Lasso"-based channel selection and "(b) LS"-based reconstruction. Another previous method REAP uses "(5) LS error"-based channel selection and "(b) LS"-based reconstruction. As shown in Table 1, POEM is better than CP and REAP in minimizing the layer-wise MSEs. Furthermore, Table 2 shows that POEM outperforms the previous methods in accuracy (both before and after fine-tuning) as well.
>
> ## I was wondering if the result trend in Table 1 could be also observable in higher layers and/or other networks.
> We added the results for more layers in Appendix C.1. The result trend was the same with we discussed in the paper: our channel selection and reconstruction approaches are better than the other ones.
>
> ## I am not sure whether omitting f’(Y) when deriving (9) from (8) is okay.
> Omitting f'(Y) is for fast calculation (at the sacrifice of optimality). As the reviewer points out, solving Eq. (9) derives the weights to minimize the LS error. In Eq. (10), we bring f'(Y) back in order to evaluate the approximate error of the outputs of the activation function. Omitting f'(Y) in Eq. (8) indeed sacrifices optimality, however, this neuron (channel) selection criteria is better than the previous ones as Table 1 and 2 indicate.
>
> ## Difference from [2]
> To be quite honest, we overlooked [2]. we added it to the related work and updated several parts of the paper accordingly. We could not do experimental comparison in this period, however, we strongly believe that POEM has an advantage, at least in neuron selection criteria. We formalize the reconstruction as Eq. (5). This makes it possible to select the neurons based on the reconstruction error, as we show in Eq. (6)-(10). On the other hand, [2] uses heuristic criteria for evaluating the importance of the neurons (which is called as "sensitivity" in [2]).
>
> ## Experiments with deeper networks (ResNet-50)
> We added experiments with ResNet-50. Please see Appendix C.5. The results suggest that POEM is better than the previous methods, however, more experiments are necessary with higher pruning ratios, and with pruning ratio optimizations, to know the winner. We are currently working on it.
>
> ## The method is introduced mainly for fully connected layers, but it would be better to (briefly) describe its extension to convolutional layers.
> We added a brief explanation of how to extend POEM to convolutional layers in Appendix B.1.
>
> ### Reference
> [2] Jiang et al., Efficient DNN Neuron Pruning by Minimizing Layer-wise Nonlinear Reconstruction Error, IJCAI'18

---

### Official Review · Reviewer_E1Gd · 2022-10-24

**Confidence:** 3
**Correctness:** 3
**Technical Novelty And Significance:** 2
**Empirical Novelty And Significance:** 3
**Recommendation:** 5

**Clarity, Quality, Novelty And Reproducibility:**

The paper is well written and is easy to follow. The proposed approach is considered the output layer after applying activation. A similar approach has been used in model distillation. It is worth extending the related work and mentioning how they are different.

**Strength And Weaknesses:**

Strength:
1- The paper motivates minimizing the output error of the activation function by illustrating that it is more important to reduce the error for the elements in non-flat zones. In flat zones of these functions, the errors are suppressed.

2- In neuron selection they use a greedy approach to pick the best neuron next.

3- Since the regression should be performed for each column separately they showed that this can be done by  parallelizing this for each j to obtain the solution efficiently

Weakness:
1- This approach looks similar to the distillation methods where the goal is to minimize the distance between the output distribution of the teacher and student model where the knowledge transfer is on feature level. The paper needs to clarify how different they are.

2- Table 1 shows the layer wise analysis of the channel selection and reconstruction methods for Conv1-1. Even though this results sounds interesting I am wondering how applying this approach in one layer would affect other layers? Also for the result in table 2 does the proposed approach apply all at once (independently) on all layers or it is an iterative approach where one is conditional on the previous pruning.

3- In both scenarios above I am wondering how the channel selection approach would affect the final result. It is true that it is shown in table 1 proposed channel selection and reconstruction is best (for one layer) but apparently the reconstruction approach improves the result independent of the channel selection. So I am wondering if you use the proposed reconstruction method but use (L1, L2, Lasso, …) for the channel selection for all the layers then what table 2 looks like? This experiment would show the importance of proposed channel selection.

4- The paper mentions that POEM is superior to the previous methods in maintaining the accuracy of the pruned model since it performs reconstruction only for the positive elements, while the previous methods perform reconstruction for all elements including negative ones. I am not sure if this is a correct statement given that this method is data dependent.


**Summary Of The Paper:**

This paper proposes a layer-wise pruning method that conducts pruning and performs reconstruction to minimize the output error caused by pruning. In reconstruction they minimize the output error of the activation function, while the previous methods minimize the error of the value before applying the activation function. Their experimental results show improvement over existing methods.

**Summary Of The Review:**

The paper proposed to minimize the output error after applying the activation function. This approach is very similar to knowledge transfer at the feature level. The paper needs to clarify the similarity and mention how different they are. Also the experimental results need to be improved to show the impact of the channel selection over the existing method.

---

> ### Author Response · Authors · 2022-11-18
> **The difference of POEM and knowledge distillation; Some extra experiments are added.**
>
> We appreciate a lot for the review. Our rebuttals are as follows. We hope the rebuttal helps to solve the concerns and to increase the score.
>
> ## Relation to knowledge distillation
> Pruning (including POEM) and Knowledge Distillation (KD) are clearly different. Performing KD for "feature-level transfer" is usually not possible. KD is to train a new smaller model (student) from scratch with the final outputs (the class possibilities) of the larger model (teacher) as the guide for training. The student model generally has fewer neurons in each layer (and fewer layers in most cases), therefore, the feature dimensions of the student model and the teacher model are different. For this reason, KD is not suitable for feature-level tarnsfer.
>
> ## How pruning in Conv1-1 layer affects the deeper layers
> We added such results. Please see Appendix C.2.
>
> ## Is the pruning in a layer conditional on the previous layer?
> Yes, pruning in a layer conditional on pruning in the previous layer. Once we perform pruning (and reconstruction) in the j-th layer, the outputs X in the (j+1)-th layer is more or less affected. Therefore, we re-calculate the X to perform pruning in the (j+1)-th layer, whereas we use the original Y so that the original layer-wise outputs are preserved as well as possible. We added a breif explanation about it in Appendix B.2.
>
> ## The importance of our channel selection criteria
> We evaluated accuracy after pruning with various channel selection criteria combined with our WLS-based reconstruction. Please see Appendix C.3. The trend is that the smaller the layer-wise MSE was, the better the accuracy (before fine-tuning) was.
> Unfortunately, we cannot show the accuracy after fine-tuning, because doing fine-tuning for all these cases within this period was beyond our time and resource limitation. However, Table 2 shows part of such results. POEM outperforms REAP (that uses "(5) LS error"-based channel selection and "(b) LS"-based reconstruction in Table 1) in both the layer-wise MSEs and the accuracy before and after fine-tuning.
>
> ## Is the statement saying "POEM is better because it performs reconstruction only for the positive elements in the case of ReLU" correct?
> We are not completely sure what the reviewer means by "given that this method is data dependent", however, our answer is as follows.
> We believe it is a correct statement. The exception is that the negative element of Y turns positive due to the error. In such cases, POEM ignores the error for that element, whereas the previous methods (CP, REAP) does not ignore it. However, as we can see the results in Table 1, 2, 3, POEM could maintain the accuracy of the pruned models much better than the previous methods, which implies that the exception mentioned above is not dominant for the layer-wise errors.

---

### Official Review · Reviewer_z5BU · 2022-10-30

**Confidence:** 4
**Correctness:** 3
**Technical Novelty And Significance:** 3
**Empirical Novelty And Significance:** 3
**Recommendation:** 5

**Clarity, Quality, Novelty And Reproducibility:**

The paper is written clearly and easy to understand.

Some experiments and observations in this paper leads to different conclusion from existing paper. Making it's not clear if the "training from scratch" baselines is well set. (more details see `Strength And Weaknesses`)

**Strength And Weaknesses:**

Strength:
- This paper start from an important observation for the problem of how to select the channels for pruning, and how to adjust the values of unpruned weights after pruning.
- The proposed formulation looks solid and reasonable: the author considers both computation complexity and algorithm accuracy in the formulation and algorithm derivation.
- From the experiments, the proposed algorithm show its effectiveness in reducing the output activation L2 error, and also show that ImageNet classification accuracy is better than compared methods, with or without finetuning.

Weaknesses
- On the pruning criterion, although it makes more sense to use the layer's output after activation function than before activation function. However, why not directly use the final output as the error can be accumulated to the final layer and minimizing each layer's output will be sub-optimal. In addition, directly optimizing the final metric: accuracy or loss should be a better choice. E.g., in LeGR[1], the pruning metric is directly set as the accuracy.
- In the experiments, pruning before finetuning is significantly worse than after finetuning (e.g., table 3). It means that finetuning is very important to pruning methods. However, in the experiments, fine-tuning is only performed for 25 epochs, and the effect of different fine-tuning hyperparameters are not comprehensively studied. I wonder if better fine-tuning is conducted (e.g., better lr schedule and longer training epochs, using knowledge distillation etc), the benefits of better channel selection and weights reconstruction is less or more.
- In [2], experiments show that smaller network trained from scratch with similar or even better accuracy than channel pruning. While in this paper, table 4, training scratch has significantly worse accuracy than any pruning methods. Why the experiments in this paper cannot repro what was observed in [2]?

[1] Chin, Ting-Wu, et al. "Towards efficient model compression via learned global ranking." Proceedings of the IEEE/CVF conference on computer vision and pattern recognition. 2020.

[2] Liu, Zhuang, et al. "Rethinking the value of network pruning." arXiv preprint arXiv:1810.05270 (2018).

**Summary Of The Paper:**

This paper proposed a new pruning algorithm Pruning with Output Error Minimization (POEM), which focus on the problem of "picking which neurons for pruning", and "how to reconstruct the optimal value for unpruned weights".

The paper's main claim is using the value after activation, as the targets and output, to minimize their difference before and after pruning.
For the proposed method, pruned channel selection and reconstruction are based on this optimization criterion.

The experiments show that using POEM's channel selection and reconstruction, DNNs can have smaller reconstruction error and better classification accuracy.

**Summary Of The Review:**

The paper proposed a pruning channel selection method and show that it is better than some other channel selection methods.
The experiments comparison is not clear on two issues: 1) it's not clear how is the advantage of proposed method if better fine-tuning is conducted; 2) training from scratch baseline in this paper gives different conclusion to an existing reference.

=================================================

Thanks for the authors' response and additional experiments results. From the new results I think the gap between different pruning methods is relatively smaller.
I think it will be a good point that the proposed method is more important on datasets with insufficient samples per classes. However, in the current version of this paper, this point was not proposed as the main contribution. I would suggest the authors to dive deep on this point and show more results in terms of this observation.
Thus I keep the previous score for the paper.

---

> ### Author Response · Authors · 2022-11-18
> **We conducted extra fine-tuning.**
>
> We appreciate a lot for the review. Our rebuttals are as follows. We hope the rebuttal helps to solve the concerns and to increase the score.
>
> ## Why not use final metric?
> We do know that it is possible to set "final metric" as the criteria for pruning and reconstruction. That is actually in our future work. We are using layer-wise metric (output error after activation function) in the current step.
>
> ## Fine-tuning epochs and fine-tuning settings
> We did extra fine-tuning (100 epochs) for VGG-16 and ResNet-18 (For MobileNet, we already conducted 150 epochs). The results show that 100-epoch fine-tuning did not cancel the performance gap between POEM and the previous methods. Please see Appendix C.4 in the updated paper.
> For the fine-tuning related parameters, we used the settings that had beed used in previous works (i.e. CP, REAP, AMC papers), with some exceptions (we raised lr for VGG-16 from 1e-4 to 5e-4, because we found the accuracy improves by doing so.)
>
> ## Discrepancy from the past literature
> The reviewer mentions the discrepancy of our argument and [1]. However, we do not deny the conclusion of [1] at all, but we try to add a new insight about it. In [1], training the models (with pruned archtecture) from scratch achieves at least as high accuracy as the pruned and fine-tuned counterpart. Their conclusion is based on the experiments with the datasets that have relatively many samples per class (ImageNet, CIFAR-10, etc). With such rich datasets, training the models from scratch is not very challenging. On the other hand, our argument is about the case of using the smaller datasets. With the datasets with insufficient samples per class (e.g. CUB-200-2011), pruning and fine-tuning the model pre-trained with a larger dataset is better than training such a model from scratch.
>
> ### Reference
> [1] Liu, Zhuang, et al. Rethinking the Value of Network Pruning. ICLR2019.

---

### Decision · Program_Chairs · 2023-01-20

**Decision:**

Reject

**Justification For Why Not Higher Score:**

See meta-review.

**Justification For Why Not Lower Score:**

N/A

**Metareview: Summary, Strengths And Weaknesses:**

This paper introduces a layer-wise pruning method by minimizing the reconstruction errors of nonlinear outputs. Unlike the previous methods that compute the errors before the nonlinear function, the proposed method does it after the nonlinearity and focuses on reducing the errors in the non-saturating regions. The authors formulate their idea as a weighted least squares method and also introduce an efficient way to solve it. The proposed approach is validated with relatively shallow networks.

All the reviewers appreciated the main idea behind the paper, finding it simple and intuitive. On the other hand, they all thought it was falling slightly below the bar for ICLR acceptance. They expressed concerns about the quality of the empirical investigation, as well as coverage of related work, which can be addressed in a revision.